# A novel *in vitro* assay model developed to measure both extracellular and intracellular acetylcholine levels for screening cholinergic agents

**Ryohei Tanaka-Kanegae** [ID]*, **Koichiro Hamada**

Saga Nutraceuticals Research Institute, Otsuka Pharmaceutical Co., Ltd., Saga, Japan

* Tanaka.Ryohei@otsuka.jp

## Abstract

### Background

Cholinergic neurons utilize choline (Ch) to synthetize acetylcholine (ACh) and contain a high-affinity Ch transporter, Ch acetyltransferase (ChAT), ACh receptors, and acetylcholinesterase (AChE). As the depletion or malfunction of each component of the cholinergic system has been reported in patients with dementia, many studies have sought to evaluate whether treatment candidates affect each of the cholinergic components. The associated changes in the cholinergic components may be reflected by intra- or extra-cellular ACh levels, with an increase in extracellular ACh levels occurring following AChE inhibition. We hypothesized that increases in intracellular ACh levels can be more sensitively detected than those in extracellular ACh levels, thereby capturing subtle effects in the cholinergic components other than AChE. The objective of this study was to test this hypothesis.

### Methods

We developed an *in vitro* model to measure both extracellular and intracellular ACh levels using the human cholinergic neuroblastoma cell line, LA-N-2, which have been reported to express Ch transporter, ChAT, muscarinic ACh receptor (mAChR), and AChE. With this model, we evaluated several drug compounds and food constituents reported to improve cholinergic function through various mechanisms. In addition, we conducted western blotting to identify the subtype of mAChR that is expressed on the cell line.

### Results

Our cell-based assay system was capable of detecting increases in extracellular ACh levels induced by an AChE inhibitor at relatively high doses, as well as increases in intracellular ACh levels following the administration of lower AChE-inhibitor doses and an mAChR agonist. Moreover, increases in intracellular ACh levels were observed even after treatment with food constituents that have different mechanisms of action, such as Ch provision and ChAT activation. In addition, we revealed that LA-N-2 cells expressed mAChR M2.

**Data Availability Statement:** All relevant data are within the paper and its Supporting Information files.

**Funding:** The funder provided support in the form of salaries for authors [RT and KH], but did not have any additional role in the study design, data collection and analysis, decision to publish, or preparation of the manuscript. The specific roles of these authors are articulated in the 'author contributions' section.

**Competing interests:** I have read the journal's policy and the authors of this manuscript have the following competing interests: The authors are employees of Otsuka Pharmaceutical Co., Ltd. This does not alter our adherence to PLOS ONE policies on sharing data and materials

## Conclusion

The findings support our hypothesis and indicate that the developed assay model can broadly screen compounds from drugs to food ingredients, with varying strengths and mechanisms of action, to develop treatments for ACh-relevant phenomena, including dementia and aging-related cognitive decline.

## Introduction

Acetylcholine (ACh) is a neurotransmitter that plays crucial roles in both the central and peripheral nervous systems, and central cholinergic transmission is essential for normal cognitive processes [1]. The cholinergic system contains choline (Ch), which is the substrate for ACh synthesis; high-affinity Ch transporter, which carries Ch into cholinergic neurons; Ch acetyltransferase (ChAT), which is responsible for ACh synthesis; muscarinic and nicotinic ACh receptors (mAChRs and nAChRs, respectively); and acetylcholinesterase (AChE). The depletion or malfunction of these components has been reported in individuals experiencing cognitive decline, leading to the evaluation of whether drug candidates can affect the abundance and availability of Ch and the activity of AChE and ChAT, and elicit agonistic effects at ACh receptors [2–6]. From this perspective, numerous food constituents have been studied as therapeutic candidates, as the importance of a nutritional approach to prevent cognitive decline has become apparent [7]. Given that changes in Ch availability and activities of the individual cholinergic components may be reflected in ACh levels inside and/or outside cholinergic cells, determining whether treatment candidates increase ACh levels may represent an index when exploring cholinergic agents.

*In vivo*, brain fixation [8] and microdialysis [9] techniques are commonly used to measure ACh levels. Although ACh in the **synaptic** cleft is rapidly hydrolyzed by AChE after its transmission, brain fixation enables the quantification of intracellular and extracellular ACh, while microdialysis quantifies extracellular ACh. Considering that the main sites of ACh biosynthesis and degradation are inside and outside cholinergic cells, respectively [3], the increase in intracellular ACh levels may be detected more sensitively than that in extracellular ACh levels. We, therefore, consider that an *in vitro* model capable of measuring intracellular ACh separately from extracellular ACh could serve as a promising tool to broadly screen cholinergic agents, especially food components with mild cholinergic activities. However, no such model exists, although a substantial number of *in vitro* models have been developed focusing on single cholinergic components such as AChE, ChAT, and AChRs [5, 10–13].

We further hypothesized that the detection of increases in extracellular ACh levels could be applied to determine the activity of AChE-inhibiting drugs, whereas the detection of increases in intracellular ACh levels would be sensitive enough to determine the subtle effects elicited by food constituents via various mechanisms other than AChE inhibition. To test this hypothesis, we developed an *in vitro* model and assessed several drug compounds and food components with a known capacity to enhance cholinergic function through different mechanisms, including AChE inhibition (physostigmine [14], delphinidin [15], and black ginger extract [16]), Ch provision (glycerophosphocholine [17] and lysophosphatidylcholine (LPC) [18]), ChAT activation (luteolin [19] and nobiletin [20]), and AChR activation (muscarine and cytisine [21]).

## Materials and methods

### Reagents

Dulbecco's Modified Eagle's Medium (DMEM)/Nutrient Mixture F-12 Ham (Ham's F-12) (D8062), physostigmine, and (+)-muscarine chloride were purchased from Sigma-Aldrich (St. Louis, MO, USA). Cytisine was obtained from LKT Laboratories, Inc. (St. Paul, MN, USA). DMEM/Ham's F-12 without Ch chloride was obtained from Cell Science & Technology Inst., Inc. (Miyagi, Japan). Fetal bovine serum (FBS) was purchased from Nichirei Biosciences Inc. (Tokyo, Japan). Penicillin and streptomycin were purchased from Thermo Fisher Scientific, Inc. (Waltham, MA, USA). ACh bromide, Ch chloride, LPC from egg yolk, and luteolin were purchased from Wako Pure Chemical Industries Ltd. (Osaka, Japan). Isopropylhomocholine (IPHC) was provided by Eicom Corporation (Kyoto, Japan). Delphinidin chloride was purchased from Tokiwa Phytochemical Co., Ltd. (Chiba, Japan), and black ginger extract was obtained from Maruzen Pharmaceuticals Co., Ltd. (Hiroshima, Japan). Nobiletin was purchased from Indofine Chemical Company, Inc. (Hillsborough, NJ, USA). Other food constituents evaluated in the present study are listed in S1 Table. Lysis buffer was obtained from Bio-Rad Laboratories, Inc. (Hercules, CA, USA). Rabbit anti-mAChR M2 antibody (Ab) (ab109226) was purchased from Abcam (Cambridge, UK) and anti-rabbit Ab horseradish peroxidase (HRP)-linked IgG Ab was obtained from Cell Signaling Technology (Beverly, MA, USA). Amersham enhanced chemiluminescence (ECL) blocking agent and ECL western blotting detection reagents were purchased from Cytiva (Marlborough, MA, USA). Mouse brain tissue lysate was obtained from Abcam. All reagents and chemicals used were of reagent grade.

### Cell culture

We used the human neuroblastoma cell line LA-N-2 for the *in vitro* model as it has a cholinergic phenotype and expresses Ch transporter, ChAT, mAChR, and AChE [18, 22–27], which makes it suitable for evaluating the effects of the candidates on intracellular and extracellular ACh levels. The cell line was purchased from the **European Collection of Authenticated Cell Cultures**. DMEM/Ham's F-12 supplemented with 10% heat-inactivated FBS, penicillin (100 U/mL), and streptomycin (100 µg/mL) was used for cell culture, whereas DMEM/Ham's F-12 without Ch chloride supplemented with 2% heat-inactivated FBS was used as an assay medium. The cells were incubated at 37°C in a humidified atmosphere of 95% air and 5% $CO_2$. The number of passages of the cells used for assays ranged from 15 to 25.

### Assays

LA-N-2 cells were seeded at $4.0 \times 10^5$ cells/well in a 24-well plate and cultured until sub-confluent. During the assay, the culture medium was replaced with 1 mL of the aforementioned assay medium with the test compound dissolved in water, ethanol, or dimethyl sulfoxide (DMSO) (final concentration: 0.1% (*v/v*) solvent). Test compounds were used in the concentration range at which they did not show any cytotoxicity. After 5 h of incubation at 37°C, the medium was collected as the extracellular fraction, and the cells were rinsed with phosphate-buffered saline (PBS) and scraped with 200 µL of 0.1 M ice-cold perchloric acid to deactivate proteins that could affect Ch and ACh levels. The residual cells and wells were washed with 200 µL of perchloric acid; the rinse solution was mixed with the aforementioned lysate in perchloric acid (the total volume is 400 µL) and stored as the intracellular fraction. The collected culture medium and cell lysates were stored at -30°C until Ch and ACh measurement.

## Choline and acetylcholine measurement

We decided to employ electrochemical detection combined with high-performance liquid chromatography (HPLC) to measure Ch and ACh levels owing to its practical advantages, including high sensitivity and selectivity, rapid response, and operational simplicity [28]. We then followed a previously described quantification method [29] with minor modifications. The HPLC system consisted of a pump, column oven, electrochemical detector (ECD) (HTEC-500; Eicom Corporation), data processor (D-7000; Hitachi Ltd., Tokyo, Japan), auto-sampler (L-7200; Hitachi Ltd.), guard column (CH-GEL, 3.0 mm internal diameter (ID); Eicom Corporation), separation column (Eicompak AC-GEL, 2.0 mm ID × 150 mm; Eicom Corporation), and enzyme column immobilized with AChE and Ch oxidase (AC-ENZYM II, 1.0 mm ID × 4.0 mm; Eicom Corporation). The temperature of the column oven was maintained at 33˚C, and a platinum working electrode was used at 450 mV versus a silver (Ag)/silver chloride (AgCl) reference electrode (both electrodes from Eicom Corporation). A mobile phase containing 50 mM potassium bicarbonate, 2.5 mmol/L (M) sodium 1-decanesulfonate, and 134 μM EDTA·2Na was used under isocratic conditions at a flow rate of 150 μL/min.

The collected samples in assays were pretreated for the HPLC analysis. The cell lysates in perchloric acid were sonicated for 3 min (Bioruptor; Tosyo Denki, Kanagawa, Japan) and centrifuged (15,000 × *g*, 15 min, 4˚C). The supernatants were collected and neutralized by adding 1 M potassium bicarbonate. After adding 45 pmol IPHC as an internal standard, the supernatants were mixed with chloroform and vigorously agitated. Delipidation was completed by centrifugation (15,000 × *g*, 10 min, 4˚C) and filtration of the supernatants through a 0.2-μm polytetrafluoroethylene (PTFE) membrane (Millex-LG; Millipore Corporation, Billerica, MA, USA). The culture media were mixed with IPHC and subjected to the same delipidation process as the cell lysates. Filtration was carried out using a 5-kDa cut-off filter (**Ultrafree**[®] **MC-PLHCC;** Human Metabolome Technologies, Inc., Yamagata, Japan), and the filtrates were injected into the HPLC system. Stock solutions containing Ch and ACh were prepared using 20 mM phosphate buffer containing 20 mM EDTA·2Na, and then aliquoted and preserved at -80˚C. For each assay, the stock solution was serially diluted, and IPHC was added to each diluted solution. The ratios of Ch to IPHC and ACh to IPHC were used to construct standard curves; the quantitative range for Ch and ACh was 20 nM to 2 μM. The Ch and ACh content in the samples was recalculated as a percentage of the Ch or ACh level in the test group to that in the vehicle-treated (control) group for interexperimental comparison. For treatments that increased ACh levels, we repeated the assay at least once to confirm reproducibility. A schematic diagram of the new assay system is shown in Fig 1.

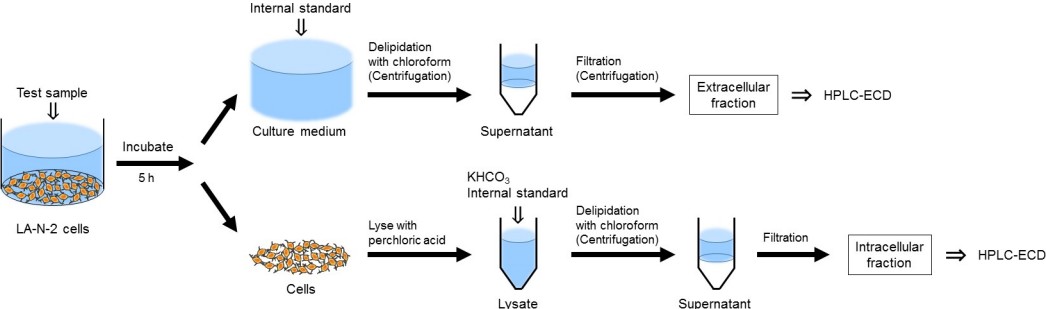

**Fig 1. A schematic diagram of the new *in vitro* assay system.** The procedures to measure extracellular and intracellular acetylcholine levels are illustrated.

## Western blotting

LA-N-2 cells ($4.0 \times 10^6$ cells) were seeded into 6-mm dishes and cultured until sub-confluent. Then, cells were treated with or without the assay medium for 5 h. Next, they were washed with PBS twice and lysed in a lysis buffer including 2 mM phenylmethylsulfonyl fluoride. The lysate was sonicated and centrifuged ($4,500 \times g$, 20 min, 4°C), and the supernatant was collected. Denatured proteins were separated by sodium dodecyl sulfate-polyacrylamide gel electrophoresis (SDS-PAGE) using 10% polyacrylamide gels, and then transferred to polyvinylidene fluoride (PVDF) membranes (Bio-Rad). After blocking with 2% Amersham ECL blocking agent in Tris-buffered saline with **Tween**-20, the membrane was treated with rabbit anti-mAChR M2 Ab (1:1000) over night at 4°C, followed by the corresponding HRP-conjugated secondary Ab (1:1000) for 2 h at room temperature. The blots were developed using ECL western blotting detection reagents. Mouse brain tissue lysate was used as a positive control.

## Statistical analysis

For each assay, significant differences between the control and test groups were evaluated using the unpaired *t*-test (two-tailed). For multiple comparisons, Dunnett's *post hoc* test was performed after one-way analysis of variance (ANOVA). Differences with probability ($p$) values < 0.05 were considered statistically significant. SAS 9.4 (SAS Institute Inc., Cary, NC, USA) was used for statistical analyses.

## Results

In LA-N-2 cells treated with physostigmine, the intracellular ACh levels increased dose-dependently, with significant differences from those of the control at doses higher than 500 nM (107.8%, $p = 0.024$ for 500 nM; 109.2%, $p = 0.009$ for 5 μM; and 113.4%, $p < 0.001$ for 50 μM), whereas the intracellular Ch levels did not change (Table 1A, Assay 1). Physostigmine at doses higher than 5 μM also led to the detection of ACh in the culture media (extracellular fraction), but not in the control group. In the control group, the extracellular Ch levels were higher than the intracellular Ch levels 5 h after replacing the culture medium with the assay medium (Table 1B, Assay 1). Subsequently, we tested delphinidin and black ginger extract. We observed that 100 μM delphinidin and 100 μg/mL black ginger extract reproducibly increased intracellular ACh levels up to 110.9% and 111.1% compared with the control, respectively. In contrast, the intracellular Ch levels did not significantly change after these treatments (Table 1A, Assays 2 and 3). Further, delphinidin and black ginger extract in the tested concentration ranges did not lead to ACh detection or any change in Ch levels in the extracellular fraction (Table 1B, Assays 2 and 3).

Next, we checked the response of LA-N-2 cells to Ch and Ch-containing compounds. After the incubation of LA-N-2 cells with 100 μM Ch, the intracellular Ch and ACh levels increased to 984.7% and 179.5% compared with the control, respectively (Table 2, Assay 1). The extracellular Ch level was above the quantitative limit, and ACh was not detected (S1 Dataset). When the cells were treated with LPC at concentrations between 3.125 and 25 μg/mL, both the intracellular Ch and ACh levels increased in a concentration-dependent manner. Ch levels reached statistical significance when treated with LPC at concentrations higher than 12.5 μg/mL (860.1%, $p = 0.014$ for 12.5 μg/mL; and 1259.7%, $p < 0.001$ for 25 μg/mL), and ACh levels at concentrations higher than 6.25 μg/mL (122.0%, $p = 0.009$ for 6.25 μg/mL; 129.8%, $p < 0.001$ for 12.5 μg/mL; and 147.6%, $p < 0.001$ for 25 μg/mL; Table 2, Assay 2). However, ACh was not detected in the extracellular fraction after LPC treatment. In addition, glycerophosphocholine

**Table 1. Changes in the choline and acetylcholine levels in LA-N-2 cells after treatment with physostigmine, delphinidin, and black ginger extract, which have been reported to have AChE-inhibiting activity.**

(A) Intracellular

| | | Ch | | | ACh | | |
|---|---|---|---|---|---|---|---|
| | | (μM) | (pmol) | (%) | (μM) | (pmol) | (%) |
| Assay 1 | Control | 0.181 ± 0.073 | 72.3 ± 29.4 | 100.0 ± 40.6 | 0.331 ± 0.018 | 132.2 ± 7.3 | 100.0 ± 5.6 |
| | 5 nM physostigmine | 0.164 ± 0.060 | 65.6 ± 24.1 | 90.7 ± 33.3 | 0.329 ± 0.008 | 131.7 ± 3.2 | 99.6 ± 2.4 |
| | 50 nM physostigmine | 0.191 ± 0.062 | 76.6 ± 24.6 | 105.8 ± 34.0 | 0.353 ± 0.007 | 141.1 ± 2.6 | 106.7 ± 2.0 |
| | 500 nM physostigmine | 0.173 ± 0.077 | 69.3 ± 30.8 | 95.8 ± 42.6 | 0.356 ± 0.003 | 142.6 ± 1.2 | 107.8 ± 0.9 * |
| | 5 μM physostigmine | 0.203 ± 0.062 | 81.2 ± 24.7 | 112.2 ± 34.1 | 0.361 ± 0.008 | 144.3 ± 3.3 | 109.2 ± 2.5 ** |
| | 50 μM physostigmine | 0.175 ± 0.050 | 70.1 ± 20.2 | 96.9 ± 27.9 | 0.375 ± 0.004 | 150.0 ± 1.6 | 113.4 ± 1.2 ** |
| Assay 2 | Control | 0.143 ± 0.039 | 57.4 ± 15.7 | 100.0 ± 27.3 | 0.389 ± 0.009 | 155.7 ± 3.5 | 100.0 ± 2.2 |
| | 25 μM delphinidin | 0.165 ± 0.070 | 66.0 ± 28.0 | 115.0 ± 48.8 | 0.415 ± 0.011 | 165.9 ± 4.6 | 106.6 ± 2.9 |
| | 50 μM delphinidin | 0.168 ± 0.055 | 67.4 ± 22.1 | 117.4 ± 38.5 | 0.415 ± 0.007 | 165.9 ± 2.6 | 106.6 ± 1.7 |
| | 100 μM delphinidin | 0.178 ± 0.085 | 71.1 ± 33.9 | 124.0 ± 59.1 | 0.432 ± 0.016 | 172.6 ± 6.2 | 110.9 ± 4.0 ** |
| Assay 3 | Control | 0.198 ± 0.092 | 79.1 ± 36.9 | 100.0 ± 46.6 | 0.452 ± 0.022 | 180.9 ± 8.9 | 100.0 ± 4.9 |
| | 25 μg/mL black ginger | 0.205 ± 0.060 | 81.9 ± 24.1 | 103.5 ± 30.5 | 0.454 ± 0.023 | 181.5 ± 9.1 | 100.3 ± 5.1 |
| | 50 μg/mL black ginger | 0.178 ± 0.068 | 71.0 ± 27.1 | 89.8 ± 34.3 | 0.458 ± 0.023 | 183.3 ± 9.2 | 101.4 ± 5.1 |
| | 100 μg/mL black ginger | 0.177 ± 0.018 | 70.6 ± 7.2 | 89.4 ± 9.1 | 0.502 ± 0.013 | 201.0 ± 5.3 | 111.1 ± 2.9 * |

(B) Extracellular

| | | Ch | | | ACh | | |
|---|---|---|---|---|---|---|---|
| | | (μM) | (pmol) | (%) | (μM) | (pmol) | (%) |
| Assay 1 | Control | 3.16 ± 0.12 | 3158 ± 123 | 100.0 ± 3.9 | ND | ND | - |
| | 5 nM physostigmine | 3.20 ± 0.08 | 3204 ± 76 | 101.4 ± 2.4 | ND | ND | - |
| | 50 nM physostigmine | 3.24 ± 0.04 | 3239 ± 45 | 102.5 ± 1.4 | ND | ND | - |
| | 500 nM physostigmine | 3.35 ± 0.19 | 3350 ± 194 | 106.1 ± 6.1 | ND | ND | - |
| | 5 μM physostigmine | 3.33 ± 0.15 | 3326 ± 155 | 105.3 ± 4.9 | 0.0310 ± 0.0012 | 31.0 ± 1.2 | - |
| | 50 μM physostigmine | 2.86 ± 0.13 | 2859 ± 126 | 90.5 ± 4.0 | 0.0339 ± 0.0026 | 33.9 ± 2.6 | - |
| Assay 2 | Control | 3.29 ± 0.11 | 3289 ± 109 | 100.0 ± 3.3 | ND | ND | - |
| | 25 μM delphinidin | 3.23 ± 0.18 | 3226 ± 181 | 98.1 ± 5.5 | ND | ND | - |
| | 50 μM delphinidin | 3.42 ± 0.08 | 3418 ± 75 | 103.9 ± 2.3 | ND | ND | - |
| | 100 μM delphinidin | 3.47 ± 0.31 | 3469 ± 306 | 105.5 ± 9.3 | ND | ND | - |
| Assay 3 | Control | 3.81 ± 0.36 | 3811 ± 356 | 100.0 ± 9.3 | ND | ND | - |
| | 25 μg/mL black ginger | 3.92 ± 0.56 | 3919 ± 565 | 102.8 ± 14.8 | ND | ND | - |
| | 50 μg/mL black ginger | 3.89 ± 0.39 | 3891 ± 391 | 102.1 ± 10.3 | ND | ND | - |
| | 100 μg/mL black ginger | 4.17 ± 0.44 | 4166 ± 442 | 109.3 ± 11.6 | ND | ND | - |

The values are expressed as absolute (μM and pmol) and relative to vehicle control (%). Data are presented as mean ± standard deviation (SD), n = 3 in all groups

$^*p < 0.05$

$^{**}p < 0.01$. ACh, acetylcholine; AChE, acetylcholinesterase; Ch, choline; ND, not detected.

and phosphatidylcholine did not increase the intracellular Ch or ACh levels; ACh was not detected in the extracellular fraction after the treatments (S1 Dataset).

Subsequently, we assessed luteolin and nobiletin using the cell-based assay system. Although extracellular ACh was not detected, intracellular ACh levels significantly increased after treatment with 50 and 100 μM luteolin (111.6%, $p = 0.006$ for 50 μM; and 128.3%, $p < 0.001$ for 100 μM), and 100 μM nobiletin (108.6%, $p = 0.025$). There were no significant changes in intracellular Ch levels (Table 3).

**Table 2. Changes in the choline and acetylcholine levels in LA-N-2 cells after treatment with choline and lysophosphatidylcholine that have a Ch moiety.**

| Intracellular | | | | | | | |
|---|---|---|---|---|---|---|---|
| | | Ch | | | ACh | | |
| | | (μM) | (pmol) | (%) | (μM) | (pmol) | (%) |
| Assay 1 | Control | 0.079 ± 0.017 | 31.6 ± 6.9 | 100.0 ± 22.0 | 0.281 ± 0.024 | 112.4 ± 9.6 | 100.0 ± 8.5 |
| | 100 μM choline | 0.778 ± 0.154 | 311.2 ± 61.7 | 984.7 ± 195.3 ** | 0.504 ± 0.012 | 201.7 ± 4.7 | 179.5 ± 4.1 ** |
| Assay 2 | Control | 0.116 ± 0.028 | 46.4 ± 11.2 | 100.0 ± 24.2 | 0.231 ± 0.013 | 92.4 ± 5.4 | 100.0 ± 5.8 |
| | 3.125 μg/mL LPC | 0.249 ± 0.102 | 99.5 ± 40.8 | 214.7 ± 87.9 | 0.265 ± 0.008 | 105.8 ± 3.2 | 114.6 ± 3.5 |
| | 6.25 μg/mL LPC | 0.443 ± 0.236 | 177.0 ± 94.5 | 381.9 ± 203.8 | 0.282 ± 0.015 | 112.7 ± 5.8 | 122.0 ± 6.3 ** |
| | 12.5 μg/mL LPC | 0.997 ± 0.366 | 398.7 ± 146.5 | 860.1 ± 316.1 * | 0.300 ± 0.015 | 119.9 ± 5.8 | 129.8 ± 6.3 ** |
| | 25 μg/mL LPC | 1.460 ± 0.694 | 583.9 ± 277.6 | 1259.7 ± 598.9 ** | 0.341 ± 0.036 | 136.3 ± 14.6 | 147.6 ± 15.8 ** |

The values are expressed as absolute (μM and pmol) and relative to vehicle control (%). Data are presented as mean ± standard deviation (SD), n = 3 in all groups

*$p < 0.05$

**$p < 0.01$. ACh, acetylcholine; Ch, choline; LPC, lysophosphatidylcholine.

Finally, we treated LA-N-2 cells with 100 μM muscarine, an mAChR agonist. Although extracellular ACh was not detected, intracellular ACh levels significantly increased up to 122.5% compared with the control ($p = 0.038$); intracellular Ch levels did not change after muscarine treatment (Table 4). Other experiments showed that treatment with up to 5 μM cytisine, an nAChR agonist, neither changed ACh nor Ch levels (S1 Dataset). Moreover, western blotting revealed that LA-N-2 cells expressed mAChR M2, even after treatment with the assay medium for 5 h (Fig 2).

We have listed drug compounds and food constituents that increased ACh levels in our cell-based assay in Table 5 with their reported mechanisms of action on the cholinergic system, and the other drug compounds and food constituents, which did not, in Table 6.

## Discussion

When LA-N-2 cells were incubated in the assay medium for 5 h as the control, the concentration of extracellular Ch was quantified to be 3–4 μM (Table 1B). Tucek reported that the concentration of extracellular Ch in the human brain is similar to that in the cerebrospinal fluid, within the range of 0.4–5.1 μM, and this notion has well been supported by microdialysis

**Table 3. Changes in the choline and acetylcholine levels in LA-N-2 cells after treatment with luteolin and nobiletin, which have been reported to have ChAT-increasing activity.**

| Intracellular | | | | | | | |
|---|---|---|---|---|---|---|---|
| | | Ch | | | ACh | | |
| | | (μM) | (pmol) | (%) | (μM) | (pmol) | (%) |
| Assay 1 | Control | 0.202 ± 0.073 | 80.9 ± 29.2 | 100.0 ± 36.1 | 0.450 ± 0.003 | 180.1 ± 1.2 | 100.0 ± 0.7 |
| | 25 μM luteolin | 0.207 ± 0.084 | 82.6 ± 33.5 | 102.1 ± 41.5 | 0.448 ± 0.001 | 179.0 ± 0.5 | 99.4 ± 0.3 |
| | 50 μM luteolin | 0.211 ± 0.102 | 84.5 ± 40.8 | 104.4 ± 50.4 | 0.502 ± 0.027 | 201.0 ± 10.9 | 111.6 ± 6.0 |
| | 100 μM luteolin | 0.223 ± 0.021 | 89.1 ± 8.4 | 110.2 ± 10.3 | 0.578 ± 0.009 | 231.1 ± 3.6 | 128.3 ± 2.0 ** |
| Assay 2 | Control | 0.143 ± 0.053 | 57.1 ± 21.3 | 100.0 ± 37.3 | 0.515 ± 0.014 | 206.1 ± 5.7 | 100.0 ± 2.8 |
| | 100 μM nobiletin | 0.113 ± 0.014 | 45.1 ± 5.5 | 78.9 ± 9.7 | 0.560 ± 0.017 | 223.9 ± 6.7 | 108.6 ± 3.3 * |

The values are expressed as absolute (μM and pmol) and relative to vehicle control (%). Data are presented as mean ± standard deviation (SD), n = 3 in all groups

*$p < 0.05$

**$p < 0.01$. ACh, acetylcholine; Ch, choline; ChAT, choline acetyltransferase.

**Table 4. Changes in the choline and acetylcholine levels in LA-N-2 cells after treatment with muscarine, a muscarinic acetylcholine receptor agonist.**

| Intracellular | | | | | | |
|---|---|---|---|---|---|---|
| | Ch | | | ACh | | |
| | (μM) | (pmol) | (%) | (μM) | (pmol) | (%) |
| Control | 0.0652 ± 0.0196 | 26.1 ± 7.8 | 100.0 ± 30.0 | 0.184 ± 0.023 | 73.6 ± 9.4 | 100.0 ± 12.8 |
| 100 μM muscarine | 0.0617 ± 0.0153 | 24.7 ± 6.1 | 94.6 ± 23.5 | 0.225 ± 0.001 | 90.1 ± 0.2 | 122.5 ± 0.3 * |

The values are expressed as absolute (μM and pmol) and relative to vehicle control (%). Data are presented as mean ± standard deviation (SD), n = 3 in all groups

*$p < 0.05$. ACh, acetylcholine; Ch, choline.

studies [30, 31]. Importantly, the concentration of extracellular Ch observed in our study fell within this range, which indicates the ability of this system to simulate the cholinergic metabolic system in physiological conditions.

ACh was not detected in the extracellular fraction in the control group (Table 1B). This supports the finding that this cell type expresses AChE [24]. Although the cholinergic phenotype of LA-N-2 cells is known to be enhanced when differentiated [24, 26], the cell line without differentiation seemed to produce a sufficient amount of ChAT for intracellular ACh to be produced and AChE for extracellular ACh to be degraded. Therefore, we prioritized a quick screen system set up and decided to perform assays without differentiation.

When LA-N-2 cells were treated with physostigmine, a commonly used AChE inhibitor, at doses higher than 5 μM, extracellular ACh was successfully detected. Given its short half-life [14], it could be possible to use an even lower concentration of physostigmine to detect extracellular ACh if the incubation time is optimized. However, 5 μM physostigmine is commonly used when performing microdialysis to increase the basal ACh level up to a detectable magnitude [32]. In addition, Lau et al. reported that their *in vitro* model using lung cancer cells could detect the increase in extracellular ACh levels after neostigmine treatment. Given that they

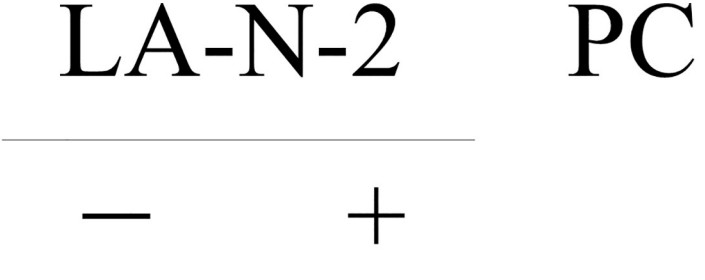

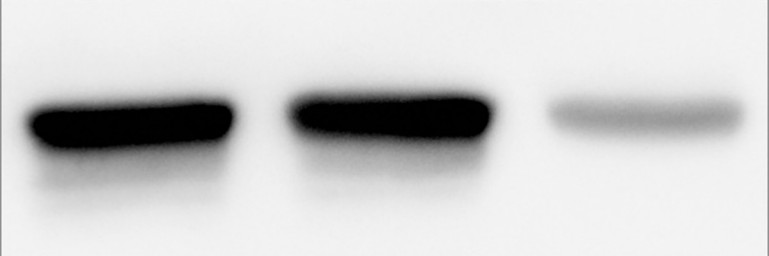

**Fig 2. The expression of muscarinic acetylcholine receptor M2 on LA-N-2 cells.** LA-N-2 cells were lysed before (-) or after (+) the treatment with the assay medium for 5 h and subjected to SDS-PAGE along with mouse brain tissue lysate as a positive control (PC). Lysates/proteins at 10 μg per lane.

**Table 5. Drug compounds and food constituents that increased extracellular and/or intracellular ACh levels in our cell-based assay.**

| Reported mechanism of action on the cholinergic system | Name | Drug/food | Concentration used in assay | Fraction where the increase was detected |
|---|---|---|---|---|
| AChE inhibition | Physostigmine | Drug | 5 nM–50 μM | Extracellular and intracellular |
| | Delphinidin chloride | Food | 25–100 μM | Intracellular |
| | Black ginger extract | Food | 25–100 μg/mL | Intracellular |
| Ch (ACh precursor) provision | Ch chloride | Food | 100 μM | Intracellular |
| | LPC | Food | 3.125–25 μg/mL | Intracellular |
| ChAT activation | Luteolin | Food | 25–100 μM | Intracellular |
| | Nobiletin | Food | 100 μM | Intracellular |
| mAChR activation | (+)-Muscarine chloride | Drug | 100 μM | Intracellular |

Abbreviations: ACh, acetylcholine; AChE, acetylcholinesterase; Ch, choline; ChAT, choline acetyltransferase; LPC, lysophosphatidylcholine; mAChR, muscarinic acetylcholine receptor.

successfully detected this increase using 50 μM neostigmine but not 20 μM [33], and that the half-maximal inhibitory concentration of neostigmine toward AChE is similar to that of physostigmine [32], we consider that our model can sensitively detect the extracellular ACh following AChE inhibition. Interestingly, using physostigmine, the detection of extracellular

**Table 6. A drug compound and food constituents that did not increase extracellular or intracellular ACh levels in our cell-based assay.**

| Name | Drug/food | Concentration used in assay |
|---|---|---|
| Cytisine | Drug | 50 nM–5 μM |
| Glycerophosphocholine | Food | 100 μg/mL |
| Arachidonic acid | Food | 20 μg/mL |
| Astaxanthin | Food | 100 μM |
| L-Carnitine | Food | 100 μM |
| Citrulline | Food | 100 μM |
| Curcumin | Food | 20 μM |
| Cyanidin chloride | Food | 100 μM |
| Cyanocobalamin | Food | 100 μM |
| Docosahexaenoic acid | Food | 20 μg/mL |
| Eicosapentaenoic acid | Food | 20 μg/mL |
| Ferulic acid | Food | 100 μM |
| Glycine | Food | 100 μM |
| Lutein | Food | 100 μM |
| Methylcobalamin | Food | 100 μM |
| Octanoic acid | Food | 100 μg/mL |
| Phosphatidylcholine from egg yolk | Food | 100 μg/mL |
| Phosphatidylethanolamine, dimyristoyl | Food | 100 μg/mL |
| Phosphatidylethanolamine, dioleoyl | Food | 100 μg/mL |
| Phosphatidylethanolamine, dipalmitoyl | Food | 100 μg/mL |
| Phosphatidylethanolamine, distearoyl | Food | 100 μg/mL |
| Phosphatidylserine | Food | 100 μg/mL |
| Pyrroloquinoline quinone | Food | 100 μM |
| L-Serine | Food | 100 μM |
| cis-15-Tetracosenoic acid | Food | 100 μg/mL |
| Zeaxanthin | Food | 100 μM |

ACh coincided with the increase in intracellular ACh levels; this increase was observed at a ten times lower dose of physostigmine (Table 1, Assay 1).

Delphinidin, an anthocyanidin commonly present in pigmented fruits and vegetables [15], and black ginger (*Kaempferia parviflora*) extract have also been reported to have an inhibitory effect on AChE [15, 16]. Treatments with these two food components increased intracellular ACh levels, although ACh was not detected in the extracellular fraction (Table 1, Assays 2 and 3). These findings support the possibility that the increase in intracellular ACh levels is more easily detected than that in extracellular ACh levels. Moreover, the findings suggest that our cell-based assay model is sensitive enough to detect the increase in intracellular ACh levels induced by food components. Given that delphinidin and black ginger extract improve cognitive function *in vivo* [15, 34], an increase in the intracellular ACh levels may serve as an indicator of cognitive improvement. For these reasons, we decided to proceed with the assays with a focus on intracellular ACh.

When Ch was added to the assay medium, it was taken up and converted to ACh in the cells (Table 2, Assay 1); this supports the existence of Ch transporter and ChAT in LA-N-2 cells. LPC, which constitutes enzyme-modified lecithin and is used as a food emulsifier owing to its amphipathic nature [35], has been shown to be taken up by and converted to ACh via Ch in LA-N-2 cells [18]. This was confirmed with 3.125 to 25 μg/mL LPC in our experimental model, and 25 μg/mL LPC evoked higher intracellular Ch levels than 100 μM Ch (Table 2, Assays 1 and 2). This finding can be explained by the intermediate-affinity uptake of Ch by LA-N-2 cells [27] and the detergent-like property of LPC; 50 μg/mL LPC caused cytotoxicity and decreased intracellular ACh levels (S1 Dataset).

Glycerophosphocholine and phosphatidylcholine caused no significant changes in intracellular Ch or ACh levels (Table 6, S1 Dataset). As LA-N-2 cells have been reported to have glycerophosphodiester phosphodiesterase activity [24], which hydrolyzes glycerophosphocholine to Ch, it seems that glycerophosphocholine was not incorporated into the cells nor degraded by the enzyme. Moreover, LA-N-2 cells have been shown to express phospholipase D to utilize phosphatidylcholine in the cell membrane [18]. However, as the cell membrane was impermeable to the phosphatidylcholine in the medium, its utilization as an ACh precursor was prevented. Similarly, we assume that phosphatidylserine, which has been shown to reverse ACh levels in aged animals [36], failed to increase ACh levels in our cell-based assay (Table 6, S1 Dataset) owing to the lack of related metabolic enzymes and its inability to cross the cell membrane from the medium.

The intracellular ACh levels increased after treatment with luteolin and nobiletin (Table 3). Luteolin is a flavonoid that is present in fruits and vegetables such as celery, chrysanthemum flower, sweet bell pepper, carrot, onion leaf, broccoli, and parsley [37]; the oral administration of luteolin has been reported to increase ACh levels in the brain of amyloid β-infused rats through ChAT activation [19]. Nobiletin, a citrus flavonoid known to improve cognitive function *in vivo* [38], has also been reported to increase ACh levels via the upregulation of ChAT expression [20]. These findings suggest that the increase in intracellular ACh levels observed after treatments with these flavonoids resulted from the upregulation of ChAT.

The cell line showed a significant increase in intracellular ACh levels when incubated with muscarine (Table 4). To the best of our knowledge, previous studies have suggested the existence of mAChR on LA-N-2 cells but have not identified the subtype expressed [25]. Therefore, we performed western blotting and showed the constant expression of mAChR M2 on the cells (Fig 2). The increase in intracellular ACh levels following muscarine stimulation may be partly mediated by mAChR M2 because the activation of this subtype leads to the feedback inhibition of ACh release and accumulation of ACh in neurons [39]. Importantly, our result demonstrated that an increase in the intracellular ACh levels owing to stimulation by mAChR

agonists could be detected by our assay system. In contrast, another ACh receptor type, nAChR, may not be expressed on LA-N-2 cells, which was supported by our data; cytisine, a nicotinic agonist [21], did not alter ACh or Ch levels (Table 6, S1 Dataset).

As shown above, our cell-based assay system could detect an increase in extracellular ACh levels by an AChE-inhibiting drug at relatively high doses. In addition, the developed model could detect an increase in intracellular ACh levels by lower doses of the AChE-inhibiting drug and an mAChR agonist, and the increase was observed even when treated with food constituents that have different mechanisms of action, such as Ch provision and ChAT activation. We consider that compared with previously reported *in vitro* models that evaluate a single component of the cholinergic system, the developed model makes it possible to more broadly screen lead compounds that have different strengths of activity and mechanisms of action by evaluating both extracellular and intracellular ACh. Although the development of cholinergic drugs has been highly focused on ACh inhibition and many AChE inhibitors have been developed [10, 11, 40], food constituents that are expected to counteract cognitive decline have various mechanisms besides AChE inhibition, as shown above [15, 16, 19, 20, 34, 38]. Therefore, the developed assay model may be particularly effective to screen food constituents with mild cholinergic activities.

According to the amyloid hypothesis, the accumulation of amyloid β in the brain is the primary factor driving the pathogenesis of Alzheimer's disease. Although a treatment approach based on this hypothesis is one of the most convincing approaches [41], multidirectional approaches may be needed for the management of dementia given the complex pathogenesis [42]. In addition, the interplay between amyloid β and the cholinergic system has been studied; the results showed that the activation of mAChR can shift the processing of amyloid β precursor protein toward the nonamyloidogenic pathway [3]. From these perspectives, the practical applicability of the developed assay model for screening cholinergic agents is evident.

Our study had the following limitations. First, we did not measure the activity of components of the cholinergic system (e.g., AChE) or investigate the interactions between the test molecules and cholinergic markers. Therefore, we cannot exclude the possibility that a test compound may have different mechanisms of action (e.g., inhibiting AChE and activating ChAT) simultaneously. Second, we used doses of drugs and food constituents up to 100 μM or 100 μg/mL because our main objective was to compare the response of intracellular and extracellular ACh. Given the metabolic process and existence of the blood–brain barrier, it is unlikely for some drugs and food constituents evaluated in this study to reach the brain at such doses [43]. The assay model described here should serve as a preliminary *in vitro* screen, and further *in vivo* and clinical studies are needed to clarify whether the candidates selected using this screening model could improve ACh transmission, neural plasticity, and even cognitive function.

In conclusion, this is the first study to report an assay model using LA-N-2 cells where both extracellular and intracellular ACh can be quantified. The simple and quick method enabled us to screen cholinergic agents, and the values were comparable and reproducible between assays when expressed as levels relative to those of the control. Notably, the increase in ACh levels by food constituents with different mechanisms of action on the cholinergic system was sensitively detected in the intracellular fraction. We propose that the developed screening system can be used in the first step of the development of new functional foods, as well as drugs, to treat ACh-relevant phenomena such as dementia and aging-related cognitive decline.

## Supporting information

**S1 Table. List of drug compounds and food constituents evaluated in our cell-based assay.** (DOCX)

**S1 Dataset. A dataset for choline and acetylcholine quantification.**
(XLSX)

**S1 Raw image. The raw image of western blot.** Fig 1 was generated from this raw image. We evaluated different lysates of LA-N-2 cells that were harvested another day, and confirmed the reproducibility of the expression of mAChR M2.
(TIF)

**S2 Raw image. The raw image of western blot.** The same membrane as S1 Raw image but with longer exposure time to increase the intensity of bands of the molecular weight marker.
(TIF)

## Acknowledgments

We thank our colleagues, especially Mr. Kiyohiko Magata, for helpful discussions. The main author is also grateful to Dr. Koichiro Hamada for managing the project.

## Author Contributions

**Conceptualization:** Ryohei Tanaka-Kanegae.

**Data curation:** Ryohei Tanaka-Kanegae.

**Formal analysis:** Ryohei Tanaka-Kanegae.

**Investigation:** Ryohei Tanaka-Kanegae.

**Methodology:** Ryohei Tanaka-Kanegae.

**Project administration:** Koichiro Hamada.

**Validation:** Ryohei Tanaka-Kanegae.

**Visualization:** Ryohei Tanaka-Kanegae.

**Writing – original draft:** Ryohei Tanaka-Kanegae.

**Writing – review & editing:** Koichiro Hamada.

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
