## [Decision Letter · Decision Letter 0]

28 May 2021

PONE-D-21-08197

A novel in vitro assay model to measure both extracellular and intracellular acetylcholine levels for screening cholinergic agents

PLOS ONE

Dear Dr. Tanaka,

Thank you for submitting your manuscript to PLOS ONE. After careful consideration, we feel that it has merit but does not fully meet PLOS ONE’s publication criteria as it currently stands. Therefore, we invite you to submit a revised version of the manuscript that addresses the points raised during the review process.

In your revised manuscript you can largely disregard the comments of Reviewer 1. But please address as fully as possible the comments of the other two reviewers, especially the detailed and constructive comments of Reviewer 3.

We look forward to receiving your revised manuscript.

Kind regards,

Israel Silman

Academic Editor

PLOS ONE

Journal Requirements:

2)  Thank you for stating the following in the Financial Disclosure section :

[This work was conducted using the resources of Otsuka Pharmaceutical Co., Ltd. and

the authors received no specific funding for this work.

The company's URL is below:

https://www.otsuka.co.jp/en/]. 

We note that one or more of the authors have an affiliation to the commercial funders of this research study :

Otsuka Pharmaceutical Co., Ltd

i. Please provide an amended Funding Statement declaring this commercial affiliation, as well as a statement regarding the Role of Funders in your study. If the funding organization did not play a role in the study design, data collection and analysis, decision to publish, or preparation of the manuscript and only provided financial support in the form of authors' salaries and/or research materials, please review your statements relating to the author contributions, and ensure you have specifically and accurately indicated the role(s) that these authors had in your study. You can update author roles in the Author Contributions section of the online submission form.

ii. Please also provide an updated Competing Interests Statement declaring this commercial affiliation along with any other relevant declarations relating to employment, consultancy, patents, products in development, or marketed products, etc.  

3) Please amend either the title on the online submission form (via Edit Submission) or the title in the manuscript so that they are identical.

4) We note that you have included the phrase “data not shown” in your manuscript. Unfortunately, this does not meet our data sharing requirements. PLOS does not permit references to inaccessible data. We require that authors provide all relevant data within the paper, Supporting Information files, or in an acceptable, public repository. Please add a citation to support this phrase or upload the data that corresponds with these findings to a stable repository (such as Figshare or Dryad) and provide and URLs, DOIs, or accession numbers that may be used to access these data. Or, if the data are not a core part of the research being presented in your study, we ask that you remove the phrase that refers to these data.

5) We note that you have indicated that data from this study are available upon request. PLOS only allows data to be available upon request if there are legal or ethical restrictions on sharing data publicly. For more information on unacceptable data access restrictions, please see http://journals.plos.org/plosone/s/data-availability#loc-unacceptable-data-access-restrictions.

Reviewers' comments:

Reviewer's Responses to Questions

**Comments to the Author**

1. Is the manuscript technically sound, and do the data support the conclusions?

Reviewer #1: Yes

Reviewer #2: Yes

Reviewer #3: Yes

2. Has the statistical analysis been performed appropriately and rigorously? 

Reviewer #1: Yes

Reviewer #2: Yes

Reviewer #3: Yes

3. Have the authors made all data underlying the findings in their manuscript fully available?

Reviewer #1: Yes

Reviewer #2: Yes

Reviewer #3: Yes

4. Is the manuscript presented in an intelligible fashion and written in standard English?

Reviewer #1: No

Reviewer #2: Yes

Reviewer #3: Yes

5. Review Comments to the Author

Reviewer #1: The manuscript, A novel in vitro assay model to measure both extracellular and intracellular acetylcholine levels for screening cholinergic agents, represents only one of many attempts to measure Acetylcholine and other cholinergic parameters as means to counter cognitive disorders improving cholinergic neurotransmission.

This model per se may be interesting but, unfortunately, ACh increasing strategy both by cholinergic agonists and by AChE inhibitors looks disastrous. Probably, we should study this system yet but its use as a therapeutic mean to fight neurodegenerative diseases as AD or other disorders, which hit cognitive function doesn't represent a priority!

Therefore, I think this manuscript has some problems with its title. In my opinion, the authors don't present a novel approach, probably the scientific community may not be interested in the measurement of cholinergic agents because they are not useful to successfully treat patients. Besides, honestly, I think we should find other means to fight beta-amyloid and neurofibrillary tangles. Later, we can use also cholinergic enhancers to improve cholinergic function.

Probably, in the meantime, we should also study better other molecules that improve significantly cognitive function in patients with mild cognitive impairment. Here you find some interesting papers that may spark your future researches.

[Choline-containing phospholipids: relevance to brain functional pathways. Clin Chem Lab Med. 2013 Mar 1;51(3):513-21. doi: 10.1515/cclm-2012-0559; Choline-Containing Phospholipids: Structure-Activity Relationships Versus Therapeutic Applications. Curr Med Chem. 2015;22(38):4328-40. doi:10.2174/0929867322666151029104152; Choline alphoscerate (alpha-glyceryl-phosphoryl-choline) an old choline- containing phospholipid with a still interesting profile as cognition enhancing agent. Curr Alzheimer Res. 2013 Dec;10(10):1070-9. doi: 10.2174/15672050113106660173; Volume Analysis of Brain Cognitive Areas in Alzheimer's Disease: Interim 3-Year Results from the ASCOMALVA Trial. J Alzheimers Dis. 2020;76(1):317-329. doi: 10.3233/JAD-190623].

Reviewer #2: ACh is an important neurotransmitter involved in a wide range of physiological processes, and the malfunction of ACh system is associated with many brain disorders. This manuscript established an in vitro model, which used human-derived cholinergic cells LA-N-2 that express several necessary cholinergic components, to measure ACh levels. Combined with HPLC to quantify the extracellular and intracellular ACh levels, such cell line is used to evaluate the effects of some drugs and food constituents on cholinergic function. The authors successfully detected the increased ACh levels affected by some drugs or food constituents. This assay is relatively easy to perform and may be useful for the preliminary screening of drugs or food constituents to treat ACh-related disorders.

It would be good that authors address the below minor critiques before their work to be published.

1. The author could draw a schematic diagram in one figure to illustrate (the principe of) this new assay. This would help readers to understand their manuscript.

2 Many food constituents have been tested in this assay under specific doses. Could authors discuss or mention whether relevant doses are similar to that of normal intake by human?

3. Based on the data you discussed, it seems that the roles of CHT1, ChAT, mAChR, and AChE cannot be distinguished when different drugs were applied (in your assay). Then, what degrees do you think the system you used could simulate the cholinergic metabolic system in physiological condition? Please discuss it.

4. I would suggest to replot at least some tables to graphs, in order for better visualization of your relevant data. There are simply too many tables.

5. In the discussion part, the authors stated, “Given that delphinidin and black ginger extract improve cognitive function in vivo [15, 28], an increase in the intracellular ACh levels could serve as an indicator of cognitive improvement.” It is simply too strong a statement in my opinion. The authors should weaken

Reviewer #3: Tanaka and Hamada use a cell model (LA-N-2 cells) that has some cholinergic properties and treat it with various compounds to screen for cholinergic effects. For this purpose, they incubated the cells with various agents (listed in Figs. 5 and S1) for five hours, then take medium as representative of the extracellular space. Subsequently, they prepared a perchloric acid extract of the cells to represent the intracellular space. Using a sensitive EICOM HPLC system, they report acetylcholine (ACh) and choline levels. They feel that increases of ACh – which were mainly detected in the intracellular space – reflect pro-cholinergic properties of the tested compounds, and that the assays may broadly reflect cholinergic stimulations because both increases of choline acetyltransferase (ChAT), or choline, or inhibition of acetylcholinesterase (AChE) are reflected in the measured values.

The methods are clearly described in this study and data are credible. In my view, however, interpretations of their findings are complicated considerations listed below.

1. LA-N-2 cells are not cholinergic by nature but they adapt some cholinergic properties (especially by up-regulation of ChAT) when differentiated e.g. by retinoic acid or LIF/CDF. It is understandable, therefore, that very little extracellular ACh is detected in most incubations because very little ACh is released by these cells, and the release machinery (SNAPs and SNAREs) may be defective. Cholinergic neurons are characterized by the expression of ChAT, CHT-1 and the vesicular ACh transporter (vAChT), and the manuscript would benefit from information (e.g. cited papers) as to the expression of their genes in LA-N-2 cells. Other cholinergic “features” listed by the authors are not exclusively cholinergic, e.g. choline is used by all cells in the body, muscarinic and nicotinic receptors are expressed by many cell types, and even AChE is expressed by several cell types, e.g. muscle cells. It would be interesting to see how the results change when the LA-N-2 cells were used in a differentiated state.

2. The measurements show intracellular choline levels of 100-200 pmol and extracellular amounts of 4,000 pmol. Unfortunately, data are not given as intra- and extracellular concentrations (µmol/L), and this should be corrected by the authors. Nevertheless, it appears that choline is concentrated much more highly in the medium, and this finding is at variance with physiological conditions: in the live brain, extracellular choline is approximately 3-4 µM whereas intracellular choline is approx. 50 µM. In vivo, this accumulation of choline within the cells is due to (a.) a negative membrane potential inside the cell which attracts the permanent cation choline and (b.) by the sodium-dependence of the high-affinity choline transporter (HACU, CHT-1), with sodium being ten times higher in the extracellular fluid. I wonder why choline is not adequately taken up by LA-N-2 cells: is there a lack of CHT-1 or of negative membrane potential ? In addition to CHT-1, choline is also transported by low-affinity transporters from the family of organic cation transporters. I wonder if these are expressed in LA-N-2 cells ? Also, where does the high choline level in the medium come from ? Fetal calf serum maybe ? Dead cells ?

3. Cholinergic activations are due to an increase of extracellular ACh. Increases of intracellular ACh do not reflect cholinergic activation, in fact they occur in vivo after administration of drugs that decrease ACh release (e.g., phenobarbitone).

4. The next surprise is the poor efficacy of physostigmine in this model. Physostigmine only enhanced ACh levels at 5-50 µM concentration, and only by 10-15%. It must be noted that physostigmine is an AChE inhibitor with a short half-life of about 20 minutes in vivo, and most of its effect may have disappeared after 5 hours of incubation. A stronger response would have been observed with an inhibitor with a longer half-life on the enzyme, e.g. rivastigmine, or with an irreversibly acting organophosphate. I also wonder whether physostigmine (at high concentrations) interfered with the detection of ACh in the HPLC which is dependent on immobilized AChE in the enzyme reactor ?

5. Nevertheless, it is noteworthy that only physostigmine, but none of the plant constituents was able to cause detectable levels of ACh in the medium. This indicates that physostigmine is able to inhibit the small amounts of AChE that may be secreted by LA-N-2 cells. As for intracellular AChE, it is known that the enzyme is processed in the Golgi apparatus and expressed together with the PRiMA anchor, but in a live neuron, the major part of AChE may not interact with ACh until secreted. I do not know how the interaction of intracellular AChE with ACh would be in a neuroblastoma cell line.

6. Choline-containing substances such as choline itself or lyso-phosphatidylcholine (lyso-PC) increased choline levels while PC and glycerophosphocholine (GPC) did not. The fact that choline at 100 µM increased intracellular choline ten-fold shows that high extracellular choline concentrations are required to increase intracellular choline – this speaks for a transport through a low-affinity carrier (see above). The fact that intracellular ACh also increased shows that intracellular choline was rate-limiting for ACh synthesis under basal conditions. It must be kept in mind that the test medium for the LA-N-2 cells was deficient in choline, a situation that does not occur in vivo – hence, addition of choline led to an immediate ACh synthesis. Before the addition of choline, the LA-N-2 cell could only use choline that was recovered from bound choline (PC mostly). To my knowledge, LA-N-2 cells are not able to synthesize choline de novo.

7. It is surprising that lyso-PC increased choline whereas GPC did not because the usual catabolic pathway of lyso-PC in the body is removal of the fatty acid by phospholipase A2 (PLA2) yielding GPC. The only explanation that comes to mind is the one that the authors give: lyso-PC may have been hydrolyzed by a PLD-like enzyme that would produce choline and monoacylglycerol (phosphate). Such PLD-like activities are usually low in the body. Maybe they are present in LA-N-2 cells ? Still, it is surprising that LA-N-2 cells do not express sufficient PLA2-activity to produce GPC and not enough phosphatase to break down GPC. It is less surprising that PC does not work because PC is rather stable, not water soluble and does not enter cells easily.

8. Muscarine increases ACh, this is surprising. Cytisin, a nicotinic agonist, does not. Which ACh receptors are expressed on LA-N-2 cells ?

9. The relevance of the data with plant constituents (e.g., flavonoids) is arguable. This reviewer does not believe that increases of ACh as observed in this study with luteolin or delphinidin would have a beneficial influence on a neurodegenerative disease such as Alzheimer´s dementia (AD). First, the extent of ACh increase is too low to affect cholinergic transmission to any measurable extent. Second, it is not known whether some of these constituents reach the blood or the brain in sufficient concentrations to exert an effect – flavonoids, for example, are extensively metabolized in the liver upon first pass. Experiments in cell culture do not reveal if substances are capable of crossing the blood-brain barrier. Taken together, it is very unlikely that delphinidin or luteolin would reach the high micromolar concentrations required to increase ACh in the brain in vivo.

6. PLOS authors have the option to publish the peer review history of their article (what does this mean?). If published, this will include your full peer review and any attached files.

Reviewer #1: No

Reviewer #2: No

Reviewer #3: No

---

## [Author Response · Author response to Decision Letter 0]

10 Aug 2021

Our response to Journal Requirements

We apologize for the insufficient formatting. We have carefully read the Journal’s style templates again and revised the format.

2-i) Please provide an amended Funding Statement declaring this commercial affiliation, as well as a statement regarding the Role of Funders in your study.

The amended Funding Statement is: “The funder provided support in the form of salaries for authors [RT and KH], but did not have any additional role in the study design, data collection and analysis, decision to publish, or preparation of the manuscript. The specific roles of these authors are articulated in the ‘author contributions’ section.”

2-ii) Please also provide an updated Competing Interests Statement declaring this commercial affiliation along with any other relevant declarations relating to employment, consultancy, patents, products in development, or marketed products, etc.

The updated Competing Interests Statement is “I have read the journal's policy and the authors of this manuscript have the following competing interests: The authors are employees of Otsuka Pharmaceutical Co., Ltd. This does not alter our adherence to PLOS ONE policies on sharing data and materials.”

3) Please amend either the title on the online submission form (via Edit Submission) or the title in the manuscript so that they are identical.

We apologize for the inconsistency. The title on the online submission form has been amended and they are now identical.

4) We note that you have included the phrase “data not shown” in your manuscript.

5) We note that you have indicated that data from this study are available upon request.

We have included all additional data in the Supporting Information and excluded the phrase “data not shown” from the manuscript.

Our response to Reviewers

Reviewer #1

Thank you for your valuable comments and for suggesting some interesting research papers related to choline-containing phospholipids. We agree that the strategy targeting beta-amyloid and neurofibrillary tangles seems to be effective. However, we consider that adopting a multidirectional approach increases the possibility to successfully cure neurodegenerative disorders, a notion that is supported by a paper you suggested, which showed the enhanced effect of donepezil when combined with choline alphoscerate. The cholinergic agents found by our in vitro system could possibly be combined with other types of treatment agents, such as anti-amyloid-beta antibody. Further research is warranted to assess the effectiveness of such a combination.

Regarding the title of our manuscript, the approach we adopted has sufficient novelty in the space of phytomedicine and method development as no other screening assays that utilized this method have been reported. Therefore, we consider the original title appropriate.

Reviewer #2

1. The author could draw a schematic diagram in one figure to illustrate (the principle of) this new assay. This would help readers to understand their manuscript.

Thank you for your helpful advice. We drew and added a schematic diagram as Fig 1.

2. Many food constituents have been tested in this assay under specific doses. Could authors discuss or mention whether relevant doses are similar to that of normal intake by human?

We consider that concentrations adopted for some food constituents may be higher than those observed in the human blood and brain after their normal intake. However, we believe that the developed model can be used for preliminary screening because it could detect the change in ACh levels after treatment with some agents, the cholinergic activities of which have already been shown in in vivo or clinical studies.

Nevertheless, this limitation has been described in the Discussion (page 25, line 379-83).

3. Based on the data you discussed, it seems that the roles of CHT1, ChAT, mAChR, and AChE cannot be distinguished when different drugs were applied (in your assay). Then, what degrees do you think the system you used could simulate the cholinergic metabolic system in physiological condition? Please discuss it.

We believe that the discrete contributions of CHT1, ChAT, mAChR, and AChE on changes in ACh levels should be addressed in another study. The concentration of extracellular Ch in the assay was maintained between 3 and 4 µM over a 5-h incubation, which is consistent with the physiological concentration.

Relevant sentences have been added in the Discussion (page 21, line 271-7).

4. I would suggest to replot at least some tables to graphs, in order for better visualization of your relevant data. There are simply too many tables.

Given the different concentration ranges of ACh and Ch, we found that it was difficult to put these values in a single graph. In addition, we have added a concentration unit (µM) considering a comment by Reviewer #3, which made it more difficult to illustrate these results in graphs. Thus, we decided to leave them as tables. We hope you are satisfied with our opinion.

5. In the discussion part, the authors stated, “Given that delphinidin and black ginger extract improve cognitive function in vivo [15, 28], an increase in the intracellular ACh levels could serve as an indicator of cognitive improvement.” It is simply too strong a statement in my opinion. The authors should weaken

We have weakened the statement (page 22, line 308).

Reviewer #3

1. LA-N-2 cells are not cholinergic by nature but they adapt some cholinergic properties (especially by up-regulation of ChAT) when differentiated e.g. by retinoic acid or LIF/CDF. It is understandable, therefore, that very little extracellular ACh is detected in most incubations because very little ACh is released by these cells, and the release machinery (SNAPs and SNAREs) may be defective. It would be interesting to see how the results change when the LA-N-2 cells were used in a differentiated state.

Cholinergic neurons are characterized by the expression of ChAT, CHT-1 and the vesicular ACh transporter (vAChT). Other cholinergic “features” listed by the authors are not exclusively cholinergic.

We appreciate your detailed and constructive comments. As you pointed out, differentiation agents enhance cholinergic characteristics of LA-N-2. However, the cell line shows a partial cholinergic phenotype by nature and have been described as cholinergic elsewhere (ref. [24, 26]). 

We also wondered if the results would change when LA-N-2 cells were differentiated. However, the cell line without differentiation seemed to produce a sufficient amount of ChAT for intracellular ACh to be produced and AChE for extracellular ACh to be degraded. Therefore, we prioritized a quick screen system set up and decided to perform assays without differentiation. This background has been added in the Discussion (page 21, line 279-84).

We apologize for the misleading description about “cholinergic” and corrected some sentences (page 2, line 11 and 22; page 4, line 40; and page 5, line 102 and 103). 

2. Unfortunately, data are not given as intra- and extracellular concentrations (µmol/L), and this should be corrected by the authors. Nevertheless, it appears that choline is concentrated much more highly in the medium, and this finding is at variance with physiological conditions: in the live brain, extracellular choline is approximately 3-4 µM whereas intracellular choline is approx. 50 µM. I wonder why choline is not adequately taken up by LA-N-2 cells: is there a lack of CHT-1 or of negative membrane potential? In addition to CHT-1, choline is also transported by low-affinity transporters from the family of organic cation transporters. I wonder if these are expressed in LA-N-2 cells? Also, where does the high choline level in the medium come from? Fetal calf serum maybe? Dead cells?

We have recalculated Ch and ACh levels and expressed them as concentrations (µM). During this process, we found a mistake in our previous calculations, and have updated the values and reconfirmed statistical significance. Although critical aspects of the results did not change, we sincerely apologize for these necessary minor changes. Given the corrected values, we can observe that the concentrations of extracellular Ch are in good agreement with those in earlier studies (ref. [29, 30]). Regarding the concentration of intracellular Ch, the cellular contents were diluted as a result of lysing with 400 µL of perchloric acid as described in the Methods. Therefore, it is difficult to compare the values obtained in this study with a previously reported one (50 µM), and to judge whether Ch was not adequately taken up. Nevertheless, given that the concentration range of extracellular Ch obtained from the assays was similar to that in physiological conditions, we assumed that the transportation system for Ch worked in our model and did not conduct further research into it. 

Although FBS contains Ch, the assay medium we used contained only 2% FBS. Therefore, we consider that the quantity of Ch derived from FBS is not significant. Dead cells were not observed under a microscope after the 5-h incubation without Ch.

3. Cholinergic activations are due to an increase of extracellular ACh. Increases of intracellular ACh do not reflect cholinergic activation.

We completely agree that the increase in intracellular ACh is not always parallel to cholinergic activation. Therefore, as a next step, an in vivo study should assess whether the treatment candidates chosen by this screening system affect cholinergic transmission.

4. The next surprise is the poor efficacy of physostigmine in this model. Physostigmine only enhanced ACh levels at 5-50 µM concentration, and only by 10-15%. It must be noted that physostigmine is an AChE inhibitor with a short half-life of about 20 minutes in vivo, and most of its effect may have disappeared after 5 hours of incubation. A stronger response would have been observed with an inhibitor with a longer half-life on the enzyme, e.g. rivastigmine, or with an irreversibly acting organophosphate. I also wonder whether physostigmine (at high concentrations) interfered with the detection of ACh in the HPLC which is dependent on immobilized AChE in the enzyme reactor?

Thank you again for your insightful comment. As a sentence has been added in the Discussion (page 21, line 288-90), 5-50 µM physostigmine is commonly used in microdialysis studies and we do not consider the efficacy of physostigmine observed in this study to be low. However, we agree that the amount of ACh detected in the extracellular fraction may increase if the incubation time is optimized for physostigmine or other AChE inhibitors with a longer half-life are applied. We consider that these possibilities should be assessed in another study.

We doubt the possibility that physostigmine interferes with AChE in the enzyme column because (a) the two columns preceding the enzyme column are packed with porous polymer, which would trap contaminants, including physostigmine in analytes, and (b) if so, we would observe an inverted concentration-response against ACh levels, which we did not. 

5. Nevertheless, it is noteworthy that only physostigmine, but none of the plant constituents was able to cause detectable levels of ACh in the medium. This indicates that physostigmine is able to inhibit the small amounts of AChE that may be secreted by LA-N-2 cells.

I do not know how the interaction of intracellular AChE with ACh would be in a neuroblastoma cell line.

Although we did not investigate the interaction of intracellular AChE with ACh in LA-N-2 cells, the activity of intracellular AChE in another neuroblastoma cell line (N18TG2) was reported by Melone et al. (Int J Dev Neurosci. 1987;5(5-6):417-28), as well as the ability of the cell to release a considerable amount of the enzyme in the culture medium.

6. The fact that choline at 100 µM increased intracellular choline ten-fold shows that high extracellular choline concentrations are required to increase intracellular choline – this speaks for a transport through a low-affinity carrier (see above). The fact that intracellular ACh also increased shows that intracellular choline was rate-limiting for ACh synthesis under basal conditions. It must be kept in mind that the test medium for the LA-N-2 cells was deficient in choline, a situation that does not occur in vivo – hence, addition of choline led to an immediate ACh synthesis. Before the addition of choline, the LA-N-2 cell could only use choline that was recovered from bound choline (PC mostly). To my knowledge, LA-N-2 cells are not able to synthesize choline de novo.

It is indeed intriguing to further explore the Ch transport system in LA-N-2 cells. However, our focus was to develop a screening system to broadly catch cholinergic agents with several mechanisms of action. Hence, we generated a Ch-deficient environment on purpose to screen agents that can be a Ch source to the cell. We consider that it is worth screening Ch-containing compounds because positive clinical results have been reported for some of them.

7. Such PLD-like activities are usually low in the body. Maybe they are present in LA-N-2 cells? Still, it is surprising that LA-N-2 cells do not express sufficient PLA2-activity to produce GPC and not enough phosphatase to break down GPC. It is less surprising that PC does not work because PC is rather stable, not water soluble and does not enter cells easily.

PLD is present in synaptosomal membranes and it has been shown that LA-N-2 cells express PLD and PLA2 (ref. [24]). In the study, the authors found that the activities of GPC-phosphodiesterases in LA-N-2 cells do not change over incubation time, which suggests that the cell line is not able to use GPC as an ACh precursor.

Related sentences have been added in the Discussion (page 22, line 317-22).

8. Muscarine increases ACh, this is surprising. Cytisin, a nicotinic agonist, does not. Which ACh receptors are expressed on LA-N-2 cells?

Given your comment, we conducted western blotting and found muscarinic AChR M2 to be expressed on LA-N-2 cells.

Related sentences have been added in the Abstract, Methods, Results, and Discussion.

9. The relevance of the data with plant constituents (e.g., flavonoids) is arguable. This reviewer does not believe that increases of ACh as observed in this study with luteolin or delphinidin would have a beneficial influence on a neurodegenerative disease such as Alzheimer´s dementia (AD). First, the extent of ACh increase is too low to affect cholinergic transmission to any measurable extent. Second, it is not known whether some of these constituents reach the blood or the brain in sufficient concentrations to exert an effect – flavonoids, for example, are extensively metabolized in the liver upon first pass. Experiments in cell culture do not reveal if substances are capable of crossing the blood-brain barrier. Taken together, it is very unlikely that delphinidin or luteolin would reach the high micromolar concentrations required to increase ACh in the brain in vivo.

As you pointed out, luteolin, delphinidin, and other flavonoids are unlikely to reach the brain at a concentration of 100 µM after normal intake. However, we consider that it is still worth evaluating these food constituents even at relatively high concentrations as a preliminary screen. As stated in our manuscript, orally administered luteolin and delphinidin have been shown to improve the cholinergic system in the brain (ref. [15, 19]). Moreover, a technique to improve the bioavailability of unstable plant constituents has been recently developed. Using this technique, curcumin has been successfully used to improve AD pathology in a clinical trial (Ma Z, et al. J Control Release. 2019), and attempts to enhance the bioavailability of luteolin and other flavonoids are being made (Ali F, et al. CNS Neurol Disord Drug Targets. 2019).

We have added sentences to discuss the doses of food constituents we used in our study (page 25, line 379-83).

---

## [Decision Letter · Decision Letter 1]

7 Sep 2021

PONE-D-21-08197R1A novel *in vitro* assay model developed to measure both extracellular and intracellular acetylcholine levels for screening cholinergic agentsPLOS ONE

Dear Dr. Tanaka,

Thank you for submitting your manuscript to PLOS ONE. After careful consideration, we feel that it has merit but does not fully meet PLOS ONE’s publication criteria as it currently stands. Therefore, we invite you to submit a revised version of the manuscript that addresses the points raised during the review process. In your revised manuscript, please address, as fully as possible, the remaining comments and criticisms of Reviewer 3.

We look forward to receiving your revised manuscript.

Kind regards,

Israel Silman

Academic Editor

PLOS ONE

Journal Requirements:

Reviewers' comments:

Reviewer's Responses to Questions

**Comments to the Author**

1. If the authors have adequately addressed your comments raised in a previous round of review and you feel that this manuscript is now acceptable for publication, you may indicate that here to bypass the “Comments to the Author” section, enter your conflict of interest statement in the “Confidential to Editor” section, and submit your "Accept" recommendation.

Reviewer #2: All comments have been addressed

Reviewer #3: (No Response)

2. Is the manuscript technically sound, and do the data support the conclusions?

Reviewer #2: Yes

Reviewer #3: Partly

3. Has the statistical analysis been performed appropriately and rigorously? 

Reviewer #2: Yes

Reviewer #3: Yes

4. Have the authors made all data underlying the findings in their manuscript fully available?

Reviewer #2: Yes

Reviewer #3: Yes

5. Is the manuscript presented in an intelligible fashion and written in standard English?

Reviewer #2: Yes

Reviewer #3: Yes

6. Review Comments to the Author

Reviewer #2: Having edited and added some results based on previous comments, the manuscript becomes clearer and more accurate. There are still some minor critiques which can be addressed by the authors.

1. The author applied many chemicals to the cell system using a series of concentrations. Will some high concentrations of the chemicals have toxicities towards cells and affect their normal metabolism? It would be better if you can discuss it in your manuscript.

2. The author mentioned that LA-N-2 cells expressed mAChR M2 using western blotting. However, the author just showed the result using the antibody of mAChR M2, probably there are more than one subtype. Discussing the expression of receptor subtypes on the membrane based on relevant RNA-seq data will be helpful.

Reviewer #3: Tanaka and Hamada have submitted a revised version of their manuscript that is clearly improved over the first version. A Western blot of m2 receptors has been added, the new Fig. 1 is useful, some values have been re-calculated, and the discussion is more to the point. Overall, my points were answered well, and several of my comments were incorporated in the revised discussion. I still think that several questions about the assay system remain, but maybe some of them can be addressed in future studies. For instance, I still do not understand (a.) why the extracellular choline level is now 3-4 µM and (b.) how the intracellular choline level can remain at 0.2 µM in this situation (Table 1), and not even increase beyond 1 µM when 100 µM choline is added to the dish (Table 2). I also do not understand how lyso-PC can evoke higher choline levels that the addition of choline itself (also Table 2). These may be the author´s measurements, but the interpretation of these findings remains a mystery to me.

In the revised version, there are some points that still need attention:

1. Line 40, the point that “the cholinergic system contains choline” is trivial. Choline is present all over the body, in blood plasma and in all cells. Please delete. It would be more reasonable to include acetylcholine as a characteristic of the cholinergic system (which was named cholinergic system out of reluctance to write “acetylcholinergic system”).

2. Line 46. Similarly, there is no evidence that lack of choline plays a role in cholinergic dysfunction or dementia, except in experimental systems. Please write “agonistic effects at/on receptors”, not “against”.

3. Line 289: The Lau et al. study does not seem to make much sense. An increase of ACh with neostigmine at 50 µM, but not at 20 µM ? Something is wrong there, in almost all labs neostigmine is active even at 1 µM.

4. Line 310: the fact that there is an increase of intracellular choline after addition of choline does not mean that there is CHT-1. Any choline transporter could do that. The authors may check the presence of CHT-1 by Western blot, or they can use hemicholinium-3 to get more information.

5. Line 316: Since there is GPC diesterase, GPC should be useable by the cells; however, the enzyme is intracellular, and GPC is not taken up by cells. A change of GPC diesterase activity is not necessary to break down GPC, its presence is enough. Please rephrase the discussion of Singh et al. so that it makes sense. Lyso-PC is a membrane detergent and may have entered the cell by unspecific mechamisms, requiring no transporters.

6. Line 317: The esterase does not “hydrate” GPC, it “hydrolyzes” GPC. Please change.

7. PLOS authors have the option to publish the peer review history of their article (what does this mean?). If published, this will include your full peer review and any attached files.

Reviewer #2: No

Reviewer #3: No

---

## [Author Response · Author response to Decision Letter 1]

24 Sep 2021

Reviewer #2

1. The author applied many chemicals to the cell system using a series of concentrations. Will some high concentrations of the chemicals have toxicities towards cells and affect their normal metabolism? It would be better if you can discuss it in your manuscript.

Thank you for your helpful advice. When the cells were treated with some chemicals including LPC at high concentrations, cell death and a decrease in ACh levels were observed. The related sentence and data have been added (page 22, line 318, and S1 dataset).

2. Discussing the expression of receptor subtypes on the membrane based on relevant RNA-seq data will be helpful.

We searched for the RNA-seq and array data regarding the expression of ACh reseptors in LA-N-2 cells but could not find any (only short RNA-seq data is available). We consider that the expression of the other subtypes of mAChR should be investigated in another study.

Reviewer #3

1. Line 40, the point that “the cholinergic system contains choline” is trivial. Choline is present all over the body, in blood plasma and in all cells. Please delete. It would be more reasonable to include acetylcholine as a characteristic of the cholinergic system (which was named cholinergic system out of reluctance to write “acetylcholinergic system”).

2. Line 46. Similarly, there is no evidence that lack of choline plays a role in cholinergic dysfunction or dementia, except in experimental systems. Please write “agonistic effects at/on receptors”, not “against”.

Although many clinical trials that administered Ch sources failed to improve the clinical status of patients with dementia, some Ch sources combined with an AChE inhibitor have succeeded and showed better outcomes than the AChE inhibitor alone (Traini E, et al. 2020, Piamonte BLC, et al. 2020). Therefore, we consider that detecting the Ch providing property of test compounds is of value and that our screening system enables it, at least for some Ch containing compounds. We consider this as one of the advantages of our system. Therefore, we did not delete the sentences you pointed out, but modified them (page 2, line 11 and page 3, line 40). In addition, we have changed the preposition from “against” to “at” (page 3, line 47).

3. Line 289: The Lau et al. study does not seem to make much sense. An increase of ACh with neostigmine at 50 µM, but not at 20 µM? Something is wrong there, in almost all labs neostigmine is active even at 1 µM.

We believe that you may be thinking of the microdialysis study with neostigmine like reviewed in [32]. In the setting, 1 µM neostigmine is enough to detect ACh because ACh is protected from AChE once it goes through the semi-permeable membrane of the microdialysis probe. However, in Lau’s and our in vitro system, ACh is subjected to hydrolysis by AChE unless the enzyme is fully inhibited or deactivated (we added perchloric acid to the medium).

4. Line 310: the fact that there is an increase of intracellular choline after addition of choline does not mean that there is CHT-1. Any choline transporter could do that. The authors may check the presence of CHT-1 by Western blot, or they can use hemicholinium-3 to get more information.

We admit that we could not clearly distinguish CHT-1 from other types of Ch transporters. According to a previous study [27], the choline-uptake mechanism of LA-N-2 involves intermediate-affinity transporters (Ch transporter-like protein 1) rather than high-affinity ones (CHT-1) (they used hemicholinium-3 in their experiments). This finding explains why a relatively low incorporation of extracellular Ch was observed in our experimental setting. The related sentence has been added to the Discussion (page 22, line 316).

5. Line 316: Since there is GPC diesterase, GPC should be useable by the cells; however, the enzyme is intracellular, and GPC is not taken up by cells. A change of GPC diesterase activity is not necessary to break down GPC, its presence is enough. Please rephrase the discussion of Singh et al. so that it makes sense. Lyso-PC is a membrane detergent and may have entered the cell by unspecific mechamisms, requiring no transporters.

We have revised the sentence (page 23, line 321). Additionally, we agree that LPC is a membrane detergent and that this property may have contributed to the higher ACh accumulation after treatment with LPC than after treatment with Ch. Related sentences have been added to the Discussion (page 22, line 317).

6. Line 317: The esterase does not “hydrate” GPC, it “hydrolyzes” GPC. Please change.

We have corrected the sentence (page 22, line 323). We appreciate all of your excellent comments and suggestions.

---

## [Editor Report · Decision Letter 2]

28 Sep 2021

A novel *in vitro* assay model developed to measure both extracellular and intracellular acetylcholine levels for screening cholinergic agents

PONE-D-21-08197R2

Dear Dr. Tanaka,

We’re pleased to inform you that your manuscript has been judged scientifically suitable for publication and will be formally accepted for publication once it meets all outstanding technical requirements.

Kind regards,

Israel Silman

Academic Editor

PLOS ONE
---

## [Editor Report · Acceptance letter]

4 Oct 2021

PONE-D-21-08197R2 

A novel *in vitro* assay model developed to measure both extracellular and intracellular acetylcholine levels for screening cholinergic agents 

Dear Dr. Tanaka-Kanegae:

I'm pleased to inform you that your manuscript has been deemed suitable for publication in PLOS ONE. Congratulations! Your manuscript is now with our production department. 

Kind regards, 

on behalf of

Prof. Israel Silman 

Academic Editor

PLOS ONE